# Evaluation of the introduction of a healthy food and drink policy in 13 community recreation centres on the healthiness and nutrient content of customer purchases and business outcomes: An observational study

**Shaan Stephanie Naughton**[1‡]*, **Helena Romaniuk**[2‡], **Anna Peeters**[1],
**Alexandra Chung**[3], **Alethea Jerebine**[4], **Liliana Orellana**[2], **Tara Boelsen-Robinson**[1]

**1** Global Obesity Centre, Institute for Health Transformation, Deakin University, Geelong, Victoria, Australia,
**2** Biostatistics Unit, Deakin University, Geelong, Victoria, Australia, **3** School of Public Health and Preventive
Medicine, Monash University, Melbourne, Victoria, Australia, **4** School of Health and Social Development,
Deakin University, Geelong, Victoria, Australia

‡ These authors are joint first authors on this work.
* Shaan.naughton@deakin.edu.au

Universität Erlangen-Nürnberg: Friedrich-
Alexander-Universitat Erlangen-Nurnberg,
GERMANY

## Abstract

### Introduction

This observational study assessed the introduction of a comprehensive healthy food and
drink policy across 13 community organisation managed aquatic and recreation centres in
Victoria, Australia, and the associated changes on business outcomes, and the healthiness
of purchases. The policy, based on state government guidelines, mandated that food and
drink availability be based on healthiness classification: 'red' (limit) <10%, and 'green' (best
choice) >50%, and the remainder 'amber' (choose carefully).

### Methods

Six years of monthly sales data were split into three periods, prior to (1/01/2013–31/12/
2014), during (1/01/2015–31/12/2016) and post (1/1/2017–31/12/2018), policy implementa-
tion. Using point-of-sale data, food and drink nutrient content, and state guidelines, items
were classified as 'red'/'amber'/'green'. Linear models with Newey West standard errors
were fitted to compare the mean value of outcomes between post- to pre-policy implementa-
tion periods, for each outcome and centre; and were pooled using random effect meta-
analyses.

### Results

Comparing post- to pre-policy implementation periods, total food sales did not change
(mean percentage difference: -3.2% (95% confidence interval (CI) -21% to 14%), though
total drink sales declined -27% (CI -37% to -17%). The mean percentage of 'red' foods sold
declined by -15% (CI -22% to -7.7%), 'amber' food sales increased 11% (CI 5.5% to 16%).

**Data Availability Statement:** Data cannot be shared publicly because of the conditions of the agreement with the commercial collaborator. For purposes of verification and replication of the study's findings, data can be accessed from the data custodian, the Associate Dean- Research, Faculty of Health, Deakin University, Australia, peter.enticott@deakin.edu.au for researchers who meet the criteria for access to confidential data.

**Funding:** This research was funded (to AP) by the Medical Research Future Fund (MRFF) Boosting Preventative Health Research initiative : Diet and chronic disease prevention: supporting implementation of priority actions in the food and nutrition system ((MRFF Boosting Prevention via the Sax Institute) (2018–2021) https://www.health.gov.au/initiatives-and-programs/medical-research-future-fund) and by the National Health and Medical Research Council (NHMRC) as a Centre of Research Excellence in Food Retail Environments for Health (RE-FRESH) ((APP1152968) https://www.nhmrc.gov.au). The funders had no role in study design, data collection and analysis, decision to publish, or preparation of the manuscript. There was no additional external funding received for this study".

**Competing interests:** SN, HR, AP, LO, and TBR are researchers within the NHMRC-funded Centre of Research Excellence in Food Retail Environments for Health (RE-FRESH) (APP1152968). SN is funded, and HR part funded, by RE-FRESH. AP is funded through an NHMRC Investigator Grant (APP1194630). AJ is funded through a Cotutelle Doctoral Studentship from Deakin University and Coventry University. TBR is funded through a Deakin University Postdoctoral Fellowship. All authors have completed the ICMJE uniform disclosure form at www.icmje.org/coi_disclosure.pdf and declare: Anna Peeters receives funding from the NHMRC and is a board member of both Obesity Australia and the Victorian Health Promotion Foundation. Alexandra Chung receives funding from the Medical Research Future Fund Preventive Health Research Initiative (1199826). Liliana Orellana has receiving consulting fees from Western Health and the World Health Organisation-Fiji through payments made to Deakin University. Alethea Jerebine was an employee of YMCA Victoria 2014-2021. Shaan Naughton, Helena Romaniuk, and Tara Boelsen-Robinson report no other funding; All authors disclose no other relationships or activities that could appear to have influenced the submitted work. This does not alter our adherence to PLOS ONE policies on sharing data and materials.

'Green' food sales did not change (3.3%, CI -1.4% to 8.0%). The mean percentage of 'red' drinks sold declined -37% (CI -43% to -31%), 'amber' and 'green' drink sales increased by 8.8% (CI 3.6% to 14%) and 28% (CI 23% to 33%), respectively. The energy density and sugar content (percentage of total weight/volume) of both food and drinks decreased.

## Conclusions

This study has shown that the implementation of a policy to improve the health of retail food environments can result in a shift towards healthier purchases. Sales revenue from foods did not decline, though revenue from drinks did, indicating future research needs to explore mitigation of this.

## Introduction

Food retail environments in Australia, like other high-income countries, are typified by the availability and promotion of highly processed, energy dense, nutrient poor, food, and drinks [1]. With approximately 1 in 4 Australian children (aged 5–14) and 2/3rds of adults experiencing overweight or obesity in 2018 [2], there is now a concerted focus on improving the health of environments where foods are advertised, purchased, and consumed [3]. One setting that has been a major target of food retail transformation internationally is community aquatic and recreation centres [4–6]. At a national level, Australian aquatic and recreation centres are estimated to be used by eight million people annually, reaping significant health benefits and contributing to urban improvement, community cohesiveness, and social inclusion [7]. Despite aquatic and recreation centres having been identified as settings that can support individuals and communities in achieving a healthier lifestyle, they are generally dominated by unhealthy food consumption due to the limited availability of healthy alternatives [5, 8, 9].

To improve the healthiness of centre food environments for customers and staff, YMCA Victoria, a community organisation that manages over 70 aquatic and recreation centres in Victoria, Australia, introduced a comprehensive healthy food and drink policy at the end of 2014 which they could start implementing from 1 January 2015 and were required to have implemented by the end of December 2016. The policy was based on the Victorian Government's voluntary 'Healthy Choices guidelines' [10], and targeted several components (product, placement, and promotion) of the traditional marketing mix to promote the consumption of healthier items and reduce the availability of unhealthier items within the centres. The policy used an interpretive traffic light system and set a target of >50% of menu items to be classified 'green' (best choice), <10% to be classified 'red'(limit), with the remaining items to be classified as 'amber' (choose carefully) [9]. A previous mixed methods study focusing on the sugar sweetened beverage (SSB) reduction aspect of the policy in centres managed by this organisation, found that 89% of customers surveyed (n = 806) supported maintenance of the policy [11]. The full introduction of this policy allowed for evaluation of its impact on business outcomes, and the healthiness and nutrient contents (both food and drink items) of customer purchases, in a natural experiment setting.

Other studies evaluating initiatives that aim to enable and direct healthier choices (by increasing the number of healthy options and restricting less healthy options) have thus far focused on auditing items for sale [6], the healthiness of the overall food environment (for instance taking into account the presence of advertising for unhealthy items) [6, 12, 13], on the removal of sugar sweetened beverages only [14], or were single site studies [15]. Prior

evaluations have commonly lacked objective sales data and long term follow up and have limited analysis of the change to the nutrient content of purchases to sugar, or have not measured this at all [5, 14–18]. To the authors' knowledge, there have been no prior evaluations of the influence of a food retail policy introduction on the nutrient content of foods sold in recreation centres [5]. Quantifying the effect of policies that change the food environment, and that can potentially impact on health and dietary behaviours, may create and support impetus for change in other settings or places, and can contribute to the prioritisation of these policies.

This study aims to address gaps in the evidence base by measuring the change to the healthiness and nutrient content of customer purchases, and business outcomes, associated with the implementation of a comprehensive healthy food and drink policy across multiple Australian (Victorian) aquatic and recreation centres.

## Methods

This study received ethical approval through the Deakin University human research ethics committee (2016–065) as a low risk study and was conducted in accordance with the World Medical Association declaration of Helsinki [19]. Patients/the public were not involved in any aspects of this study, outside of the organisation and its staff that operated the centres. Customer level sales data was not accessed, data was accessed at a centre/retailer level. Staff at centres were involved only for facilitating auditing of food retail environments and completing questionnaires related to usual suppliers of food and beverages available for sale from their retail outlet. A representative of the organisation (the YMCA Victoria general manager of Advocacy and Health Promotion at the time) was a named investigator/applicant on the ethics application and provided organisational consent.

### Study design and policy description

This is an observational study of a natural experiment where a policy was introduced by YMCA at the end of 2014, for implementation from 1/1/2015. The policy was based on the voluntary Victorian Government 'Healthy Choices guidelines' [10] and was to be implemented by the end of December 2016. The policy stipulated that of the items available for purchase <10% of items were to be 'red' classified, and >50% 'green'. Traffic light labelling of items for sale was not explicitly in the policy and was at the discretion of the individual centres. Policy implementation was phased, allowing stock of 'red' items to be sold, rather than immediately removed from sale. Some centres implemented changes quicker than others, and some temporarily reintroduced some products in response to customer feedback. At an organisational level, the changes to drinks sold was initially prioritised, followed by foods, though some centres chose to target both at the same time. Policy implementation was supported by a health promotion officer based at YMCA's head office, with each centre responsible for implementing the policy into their own food retail outlet. Centres were provided with toolkits (developed by YMCA and containing product lists of healthier choice options from existing and new suppliers, fridge layout guides, marketing and promotional materials, and audit data collection toolkits), and menu assessments to facilitate policy implementation. Annual auditing of compliance (<10% of items available classified as 'red' and >50% as 'green') was performed by each centre, with audits shared within the organisation to highlight achievements and encourage centres to reach policy targets.

### Study inclusion criteria

Centres were considered eligible for inclusion in this analysis if they were open all year round (excluding seasonal centres); had agreed to implemented the policy; sold food (fresh,

preprepared, or packaged) from a kiosk or café (e.g., excluding centres with only vending machines); provided sales and attendance data (number of people attending the centre each month) for the study period; managed by YMCA for the study duration; (e.g., the centre had not come into or left the organisation's management); had a 2018 annual food service outlet turnover over $AUD 25,000; did not undergo a refurbishment that significantly disrupted or halted the sale of food and drinks during the study period; and did not move venues.

### Sales data

Monthly point of sale data for food and drink during the study period January 2013—December 2018 was obtained from YMCA's central database. For each product sold, data included number of items sold and total sales value before application of goods and service tax (where applicable). The nutritional content of packaged and preprepared items was obtained from manufacturers/suppliers or through the Foodworks Professional database (version 9; Xyris Pty Ltd, Brisbane, Queensland, Australia). For items prepared onsite, the nutritional content was calculated from recipes (where provided) using Foodworks Professional, or a standard composition (when recipe not provided) based on staple ingredient lists supplied by each centre and serving sizes contained in the AUSNUT 2011–13 Food Measures Database [20]. Items that could not have their composition determined (e.g., 'coffee', 'kids birthday parties') were included in sales ($) analyses but could not be included in analyses where nutritional details and volume were required. Items were classified as being 'red'/'amber'/'green' using the 2016 Victorian 'Healthy choices: food and drink classification guide' [10]. Classification was based on food and drink item types, and used criteria related to serving size, energy per serve, and sugar, fat, saturated fat, salt, and fibre contents, with some categories also having guidelines around specific ingredients (and their proportion of the final item), preparation methods, or item attributes [21]. Items were classified by an accredited practising dietitian (SN, a certified or registered dietitian in Australia), with 10% cross checked by a second dietitian (AC) to ensure accuracy of classification. Dietary fibre content was not included as this is optional information on nutrition information panels in Australia and was missing from the majority of items.

### Outcomes

Outcomes were calculated monthly for each centre, separately for food and drinks. Business outcomes: total sales (pre-tax $, for all items sold including those whose nutritional content could not be determined). Healthiness of customer purchases: to estimate the amount of 'red', 'amber' and 'green' items sold, total volume (litres or kg) of items in each classification were divided by total volume (litres or kg) sold, for all items whose nutritional content was determined) and multiplied by 100 to compute the percentage of volume (litres or kg) sold of 'red', 'amber', 'green' items. Percentage sales of 'red', 'amber', 'green' items were calculated using total sales ($, for all items whose nutritional content was determined). Nutrient content of purchases: total energy density (kJ/g or ml), considered a measure of the overall nutritional quality of items sold [22, 23], was calculated using energy sold (kJ) divided by total volume (in g or ml, for all items whose nutritional content was determined). To estimate the sugar, total fat, saturated fat, and sodium content sold, the total volume of each (all in kg) was divided by total volume (litres or kg, for all items whose nutritional content was determined) to compute the percentage of total volume sold (litres or kg) of sugar, fat, saturated fat, and sodium.

## Covariates

Monthly attendance data, number of people entering centre through turnstiles (automatically counted), was obtained through YMCA's central database. Centre socioeconomic position (SEP) was based on the centre's postcode and the Australian Bureau of Statistics Socio-Economic Indexes for Areas (SEIFA), Index of Relative Socio-Economic Advantage and Disadvantage (IRSAD) state percentiles [24], with percentiles classified as being high SEP (IRSAD percentile >65, relative lack of disadvantage and greater advantage), medium (IRSAD percentile 34–65), and low (IRSAD percentile < 34, relative greater disadvantage and a lack of advantage). Type of food preparation at each centre was classified as: no facilities (no preparation facilities, limited storage (fridge), ability to heat packaged and/or preprepared items); limited facilities (ability to prepare simple items (e.g. sandwiches) and heat packaged and/or preprepared items, limited storage (fridge)); or full food preparation facilities (ability to prepare and serve a range of hot and cold meals (including to order), full storage (ambient/cold/frozen)).

## Analysis

The 6 years of data were split into three study periods, each containing 24 months, pre-implementation, representing the period prior to the start of the initiative (1 January 2013 to 31 December 2014), during implementation (1 January 2015 to 31 December 2017) and post implementation (1 January 2018–31 December 2018).

First, we estimated the change in the outcomes between post- and pre-policy implementation periods separately for each centre and outcome by fitting a linear model with Newey-West standard errors to accommodate for serial autocorrelation (time lag of 3 assumed). For each sales outcome $Y$, the model displayed in the following equation was fitted.

$$Y_{it} = \beta_{i0} + \beta_{i1}t + I(24 < t \leq 48)\beta_{i2} + +I(48 < t \leq 72)\beta_{i3} + \beta_{i4}A_{it} + \beta_{i5}M_{1,it} + \beta_{i6}M_{2,it}$$
$$+\beta_{i7}M_{3,it} + \beta_{i8}M_{4,it} + \beta_{i9}M_{5,it} + \beta_{i10}M_{6,it} + \beta_{i11}M_{7,it} + \beta_{i12}M_{8,it} + \beta_{i13}M_{10,it} + \beta_{i14}M_{11,it}$$
$$+\beta_{i15}M_{12,it} + \upsilon_{it}$$

here $Y_{it}$ represents an outcome for site $i$ ($i = 1,2,...,13$) at time $t$ ($t = 1,2,3,...,72$ months); $I(B)$ is an indicator function representing the study period taking the value 1 if condition $B$ is true and 0 otherwise with pre-implementation period $I(t \leq 24)$ used as the reference category; $A_{it}$ represents monthly centre attendance at site $i$ and time $t$; $M_{1,it}$ to $M_{12,it}$ are indicator variables for calendar month with September $M_{9,it}$ used as the reference category, while the random variable $\upsilon_{it}$ represents the residual for centre $i$ at time $t$.

Marginal means were estimated for each study period at the mean value of the covariates. For all outcomes except total food and drink sales ($), and for each centre we estimated the mean difference between post- and pre-implementation study periods. For total food and drink sales ($), we estimated the percentage difference between post and pre-implementation sales, i.e. (mean sales post–mean sales pre)/mean sales pre × 100. For the total $ sales outcomes a sensitivity analysis was performed, fitting the same model with outcomes $\log_e$-transformed; and the percentage difference reported as sympercents [25].

For each outcome, the centre level estimates were combined using a random effects meta-analysis, with z-test used to assess if the overall change was different from zero. Heterogeneity of estimates across centres was assessed by $I^2$ statistic (percentage of the between centre variability) and tested using Cochran's Q which is based on $\chi^2$ distribution. We report centre estimates (identified with code numbers to maintain anonymity) and overall estimates with 95% confidence intervals (CI) in forest plots. To check the robustness of the autocorrelation assumptions, the analysis was repeated with assumed lags of 2, 4 and 5; findings were similar

across the different lag values (results not reported). Stratified meta-analyses were performed to investigate if the estimated changes in food and drink sales ($) outcomes differed by centre characteristics: SEP; and type of food preparation facilities. We report the combined estimate with 95% CI for each stratum. With low rates of missing data (5 months of sales data in 2 centres, and 9 months of attendance data in 6 centres) and complete data required for autocorrelation, last observation carried forward was used to handle missing values. An intention to treat approach (evaluating the policy without measuring its level of implementation) was used.

The analytical approach and targets of estimation were chosen based on the following considerations: i) The change between post and pre initiative implementation was estimated at the centre level and the overall change calculated using a meta-analysis approach instead of jointly modelling each outcome for all centres, i.e., using a multilevel model. The main reason for this approach was that the seasonal sales patterns of different centres were not aligned (for example, some centres had peaks in sales in December, while others did not; see S3 Fig); therefore, a multilevel model with a unique coefficient for each calendar month failed to adequately adjust for seasonal sales pattern at centre level. ii) The most common approach for the analysis of natural experiments interrupted time series analysis (ITSA), was found to not be appropriate for this analysis. ITSA estimates counterfactual outcomes (i.e., expected outcomes had the intervention not occurred) under a set of strong assumptions. We found that the linear assumption for the pre-implementation period was systematically violated for most outcomes in some of the centres making it impossible to estimate counterfactual outcomes (see S3 Fig). iii) Due to the characteristics of the policy implementation (see *Study design and policy description*) we only report the main estimate of interest, the comparison between post- and pre-implementation study periods. The other two comparisons (during- versus pre-implementation and post-versus during- implementation) are reported briefly.

All analyses were performed in Stata 17.

## Results

Of the 40 centres managed by the organisation that introduced the comprehensive healthy food and drink policy, 35 were non-seasonal centres, of which 13 centres were eligible for inclusion (reason for exclusion, low turnover ($<$ AUD $25,000 per annum), n = 8; contract for facility management changed, n = 7; did not sell food, n = 5; underwent renovations, n = 2; moved to larger facility, n = 1 (note one centre had two reasons for exclusion)), each providing 72 months of sales. Thirteen sequential months of sales data were excluded (1.4%: first 4 months for 1 centre as it was opening up a new café, and first 9 months in another centre that did not provide complete sales data), and 5 non-sequential months in 2 different centres with partially incomplete sales data (0.5%) had their outcomes replaced with previous months' values carried forward for analysis. Eleven of the 13 centres were in metropolitan areas and two in regional areas. Four centres located in areas with low SEP, 4 with medium SEP, and 5 with high SEP, and ranged in the level of food preparation facilities (3 had no facilities; 6 had limited facilities; and 4 had full preparation facilities).

Over the six year period, the 13 centres sold 1,012 unique food and beverage items. There was good inter-rater reliability for categorisation of food items (kappa 0.83). Items that differed in classification were generally 'amber' food items that could be classified as 'red' depending on minor composition differences between brands. A total of 2,633,748 items were sold over the six year period, with 78% (2,061,728) able to be classified. Of the items that couldn't be classified 94% were coffee and other hot drinks. 78% of unique items were packaged items coded with brand specific nutrient information, with the composition of 6.3% calculated from recipes provided or standard serves. Items that had their composition estimated from point of

sale information accounted for 0.83% of the total items sold over the study period. The sales data indicates that the amount of choice varied between centres, with the total number of individual items available in each centre over the study period ranging from 110 to 467, with a median of 273.

## Change in total sales ($)

The introduction of the policy was not found to be associated with total sales ($) of food across the 13 centres (mean percentage difference: -3.2%, 95% CI -21% to 14%, p = 0.726, Fig 1a), while there was a decline in sales of drinks, mean percentage difference -27% (95% CI -37% to -17%, p<0.001, Fig 1b) between post-implementation and pre-implementation study periods. A stratified analysis by SEP and food preparation capabilities, showed that percentage changes in food and drink sales did not differ by these strata (S1 Fig). Sensitivity analysis with the outcomes log$_e$-transformed prior to modelling, found that findings were similar with sympercent difference estimates larger in magnitude, especially for drink sales (S2 Fig).

## Change in volume of 'red', 'amber', and 'green' classified items sold

The percentage volume of 'red', 'amber' and 'green' food and drinks sold over time by each centre is shown in S3 Fig. Means and 95% CI for these outcomes in each study period by centre are displayed in S4 Fig. Overall pre- and post-implementation means for the 13 centres are shown in Table 1. For 'red' food, the mean difference between post-implementation and pre-implementation sales was -15% (95% CI -22% to -7.7%, p <0.001), 'amber' food increased by 11% (95% CI 5.5% to 16%, p<0.001), and 'green' did not change (3.3%, 95% CI -1.4% to 8.0%, p = 0.166) (Table 1, S5a–S5c Fig). The mean difference in 'red' drinks sold was -37% (95% CI -43% to -31%, p<0.001); 'amber' drinks increased by 8.8% (95% CI 3.6% to 14%, p = 0.001), and 'green' drinks increased by 28% (95% CI 23% to 33%, p<0.001) (S5d–S5f Fig).

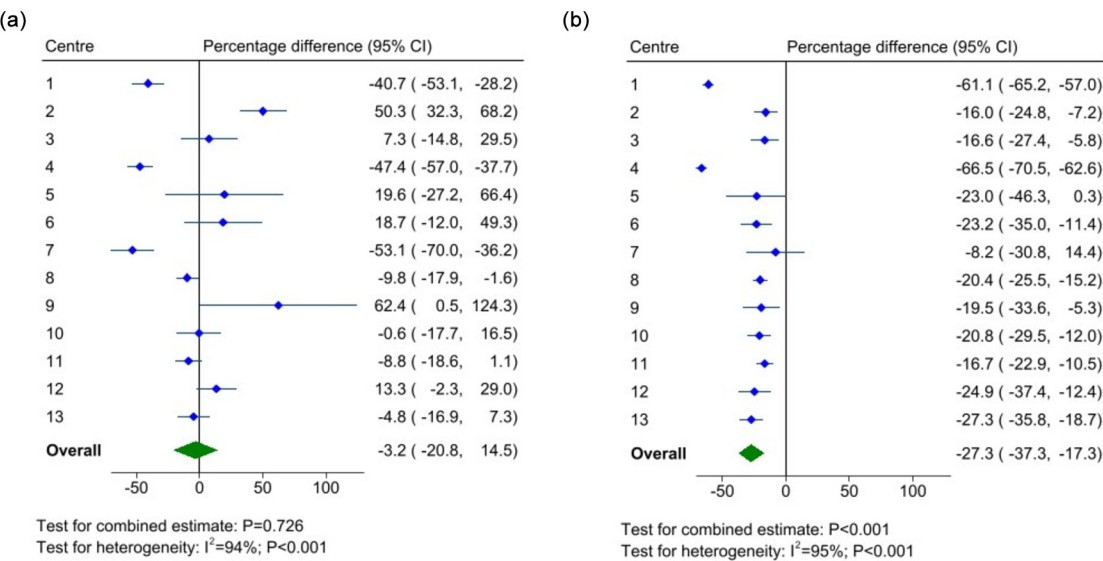

**Fig 1. Percentage difference# in total food and drink sales ($) between post and pre-implementation periods.** a) Food and b) Drink. # Percentage difference = (post–pre-intervention) / pre-intervention × 100. Overall effect estimated using a random effects REML model. CI: confidence interval.

A sensitivity analysis was performed, with percentage of sales calculated using total sales ($) rather than volume/weight. Findings were similar to when sales were calculated using volume/weight (S1 Table).

### Nutrient content of customer purchases

Means and 95% CI for nutrient sales in each study period by centre are shown in S6 Fig. The energy density of foods sold declined by -1.4 kJ/g (95% CI -2.5 kJ/g to -0.28 kJ/g, p = 0.015) between post- and pre-implementation periods (Table 1, S7a Fig). The sugar content of foods sold, as a percentage of total weight of all foods sold, decreased by -4.9% (95% CI -8.1% to -1.7%, p = 0.002) (S7b Fig). The total fat, saturated fat and sodium content of foods sold did not change (Table 1, S7c–S7e Fig).

The energy density of drinks sold declined by -0.32 kJ/ml (95% CI -0.42 kJ/ml to -0.23 kJ/ml, p<0.001) (Table 1, S7f Fig). The percentage sugar content of drinks sold declined by -2.0% (95% CI -2.5% to -1.6%, p<0.001) (S7g Fig). The total fat and saturated fat contents of drinks sold increased, and the sodium content decreased (Table 1, S7h–S7j Fig).

**Table 1. Change in healthiness of food and drink sold between post and pre-implementation periods.**

| Outcomes [#] | Pre-implementation | Post-implementation | Difference (Post–Pre) | | I² Test for Heterogeneity[&] |
|---|---|---|---|---|---|
| | Mean (95% CI)[¥] | Mean (95% CI) [¥] | Overall difference (95% CI) [¥] | p-value | |
| HEALTHINESS | | | | | |
| Food | | | | | |
| 'Red' (%) | 84 (77 to 91) | 69 (59 to 79) | -15 (-22 to -7.7) | <0.001 | 96% |
| 'Amber' (%) | 10 (6.1 to 14) | 22 (15 to 29) | 11 (5.5 to 16) | <0.001 | 97% |
| 'Green' (%) | 5.8 (1.3 to 10) | 8.9 (3.6 to 14) | 3.3 (-1.4 to 8.0) | 0.166 | 100% |
| Drinks | | | | | |
| 'Red' (%) | 45 (38 to 51) | 6.1 (3.8 to 8.3) | -37 (-43 to -31) | <0.001 | 91% |
| 'Amber' (%) | 9.6 (6.1 to 13) | 19 (13 to 24) | 8.8 (3.6 to 14) | 0.001 | 96% |
| 'Green' (%) | 45 (40 to 51) | 74 (68 to 79) | 28 (23 to 33) | <0.001 | 91% |
| NUTRIENT CONTENT | | | | | |
| Food | | | | | |
| Energy density (kJ/g) | 12 (11 to 13) | 11 (9.0 to 12) | -1.4 (-2.5 to -0.28) | 0.015 | 98% |
| Sugar (%) | 23 (18 to 27) | 18 (14 to 21) | -4.9 (-8.1 to -1.7) | 0.002 | 98% |
| Total fat (%) | 10 (8.5 to 12) | 9.4 (7.8 to 11) | -0.65 (-2.6 to 1.3) | 0.524 | 98% |
| Saturated fat (%) | 4.2 (3.6 to 4.9) | 4.0 (3.4 to 4.7) | -0.21 (-0.82 to 0.40) | 0.500 | 96% |
| Sodium (%) | 2.1 (1.7 to 2.6) | 1.9 (1.5 to 2.3) | -0.18 (-0.48 to 0.11) | 0.224 | 95% |
| Drinks | | | | | |
| Energy density (kJ/g) | 0.73 (0.62 to 0.83) | 0.41 (0.27 to 0.55) | -0.32 (-0.42 to -0.23) | <0.001 | 94% |
| Sugar (%) | 3.5 (3.0 to 4.0) | 1.5 (1.0 to 2.0) | -2.0 (-2.5 to -1.6) | <0.001 | 93% |
| Total fat (%) | 0.10 (0.029 to 0.18) | 0.17 (0.089 to 0.26) | 0.068 (0.020 to 0.12) | 0.006 | 97% |
| Saturated fat (%) | 0.064 (0.017 to 0.11) | 0.10 (0.051 to 0.16) | 0.039 (0.0063 to 0.072) | 0.020 | 97% |
| Sodium (%) | 0.20 (0.17 to 0.23) | 0.11 (0.080 to 0.14) | -0.090 (-0.12 to -0.064) | <0.001 | 94% |

[¥] Pooled meta-analysis estimates using a random effects REML model.

[#] % sales outcomes based on total volume sold

Mean centre marginal means estimated at each study period

I² estimate of the percentage of the between study variability

[&] Cochran's Q test for heterogeneity: p<0.001 for all outcomes

CI confidence interval

## Comparison of sale outcomes between other study periods

For all food outcomes, there was no change in sales between the during- and pre-implementation periods. When comparing post- and during-implementation periods, the direction and magnitude of changes for the food outcomes were essentially the same as those reported for post- vs pre-implementation periods.

For the drink outcomes, significant differences in sales were seen for both comparisons (during- and pre-implementation period, post- and during-implementation period), with the exception of the amber and saturated fat content outcomes, with changes in the same direction as the post to pre-implementation comparison for the during to post comparison. See S4 and S6 Figs as examples.

## Discussion

### Statement of principal findings

This evaluation, using six years of sales data, has shown that implementing a comprehensive healthy food and drink policy might result in a shift to healthier food and drink purchases, and a decrease in the energy density and sugar content of food and drinks purchased. In this study, policy implementation did not seem to be associated with the total value of food sales, while total drink sales value decreased by around one quarter.

### Meaning of the study: Possible explanations and implications for clinicians and policymakers

This study showed a strong decrease in the percentage of 'red' foods, and an increase in 'amber' foods sold, however, the percentage of 'green' foods did not change. This is likely due to a number of reasons, including a lack of 'green' food alternatives to popular 'red' food items, a customer preference for 'amber' foods compared to 'green' foods, and facility barriers to preparation of fresh food alternatives. The 'green' classification contains predominately freshly prepared or unprepared foods [10], and some smaller centres are challenged by this, having a combination of slower stock turnover due to attendance rates and limited preparation facilities, potentially restricting the ability of centres to prepare, store, and turnover 'green' foods without significant waste. Centres with limited food preparation ability need to rely on packaged and preprepared items, which may be limited to the range that is available through local suppliers, issues which have been found in previous research [26].

This study found a small, though significant decline in the energy density of both food and drinks sold. Additionally, there was a decrease in the percentage of sugar in both food (-4.94%) and drinks (-2.01%) sold. These decreases, taken with the reduced percentage of total sales being 'red' items, might indicate a shift towards overall purchases being healthier than before the policy was implemented. There is evidence that by increasing the healthiness of foods and drinks available for purchase, the need for individual choice and motivation towards consuming a healthier option is reduced, promoting a shift to healthier purchasing patterns [27].

This analysis found small, though significant, increases in the fat and saturated fat contents of drinks sold. Though not explored here, this is likely due to the guidelines promoting a shift from water based drinks to reduced fat dairy based drinks, due to their general health benefits [28].

In this study revenue from food sales was maintained when the availability of healthier options was increased, which to the authors' knowledge has not previously been investigated at a whole of outlet level. This finding supports the use of comprehensive healthy food and drink policies, over policies that only change the healthiness of drinks for sale (which are more

common), which also limit the ability of customers to substitute purchases of unhealthy drinks for foods. In contrast to food sales, drink sales declined post implementation. In this study we were not able to identify the extent to which customers consumed more free drinking water, brought drinks into the centre, or purchased cheaper drinks (e.g., bottled water). It should also be noted that this is total sales value, and therefore is not a proxy for profit, as the wholesale cost of items was not available for analysis.

This evaluation adds to the evidence of the efficacy of initiatives to improve the health of food environments, in real world community settings, which is relevant for policy makers considering implementing similar policies in comparable settings. By analysing six years of continuous sales data, this evaluation provides longer term outcomes than previous analyses of policies based on the Victorian Government 'Healthy Choices guidelines' [14] whilst also capturing variations in items sold [6]. The inclusion of centres from a range of locations (metropolitan and regional), food preparation capabilities, and SEP mean that the findings will be broadly relevant to other centres and policy makers looking to implement and/or regulate healthy food environments within their purview. Moreover, addressing healthy diets through an upstream policy approach reaches a wider proportion of the population than targeted approaches addressing the behaviours of specific groups of individuals, and also requires less action from the community [29].

## Strengths and weaknesses of the study

The main strength of this study is that it comprised of a wide range of objectively measured outcomes, including nutrition and business outcomes, across a 6-year period. Long-term sales data, compared to cross sectional sampling; objective measures of purchases, compared to self-reported purchases [30] and moving beyond measures of food availability to assess whether a change in availability is associated with purchasing (revenue, volume, and nutrition composition) are key strengths. Evidence from this observational study suggests that such a policy can produce changes on this range of outcomes, essential evidence for long term business function and the sustainability of a healthy food policy. The inclusion of centres situated in metropolitan and urban areas, of varied socioeconomic advantage, and differing types of food preparation capabilities, provides evidence representative of the broader community landscape. The outcomes of this study contribute to the limited evidence of the effectiveness of initiatives in promoting healthy food and drink consumption in general, and specifically in aquatic and recreation environments.

The primary weakness of this study is its observational nature, lack of control centres, and the inability to account for the effect of other unmeasured factors including secular trends in sales or prices, and other initiatives implemented in Victoria or Australia. A control group could not be included as all centres in the state were required to implement the policy. Though there were a number of facilities that failed to implement the policy, these could not be considered controls as there was an extensive campaign within the organisation to improve food and drink healthiness that might have had an effect on every centre, and also as the centres that didn't adhere to the policy are a self-selected sample and are therefore not true controls. One factor which may have influenced the results of this study is that some centres chose to display traffic light labelling on the items available for sale, which may have further influenced customer choice [31]. Other healthy eating policies in place at the time may have influenced secular trends, though healthy food policy progress was limited in Victoria [32], and federally [33], at the time. No other policies relating to sports and recreation settings were enacted during this period, with this organisation being the first to introduce the voluntary policy across all of their centres [9]. At a state level, the only policy to be introduced was the introduction of the

same voluntary policy in schools, childcare, and workplaces [32]. The only other policy which may have had an influence on the consumption of healthier food was the introduction of the Health Star Rating system in 2014, though an evaluation of the system, in October 2018, found that population awareness of the scheme was low, and that only 20–28% of eligible products adopted the rating system, with this being skewed towards items that received a higher rating (higher rating indicates a healthier alternative when comparing like products) [34]. Also, as can be seen from the outcomes plotted over time, changes to sales occurred at different times as the policy was implemented, decreasing the likelihood of external extraneous factors being the primary factor influencing results. Another limitation of this study was the number of centres eligible for inclusion (37% of non-seasonal centres) in the analysis. Though centre eligibility may have been influenced by the length of the study, investigation of long-term outcomes was key. The change to the salt content of food purchases may also have been underestimated, as several 'red' items that were removed from sale often have additional salt added after cooking (e.g., deep fried foods), which is not included in the nutritional information. Finally, not all items sold could be classified, with the majority of items excluded from the volume and nutritional analysis being hot beverages.

## Strengths and weaknesses in relation to other studies, discussing important differences in results

In comparison to similar studies evaluating the introduction of healthy food and drink policies, this study provides a more thorough analysis, by evaluating not only the volume change of different food and drink classifications sold, but also the change to the energy density, and nutritional composition of purchases [5]. Changes to all nutrients focused on in the classification system were able to be estimated, with the exception of dietary fibre, as well as changes to the energy density of items sold, which has not (to the authors' knowledge) been measured in previous studies. As highlighted in a recent systematic review of food service initiatives (in general settings), nutritional outcomes of initiatives that aim to enable healthier diets have not been measured [16]. In aquatic and recreation settings several previous studies have focused primarily on how the healthiness of available food and drinks meet policy requirements, without investigating changes to purchases [6, 13, 35, 36]. Studies that have included purchase data indicate that purchasing patterns reflect the availability of items for sale, an assertion that we believe our outcomes strengthen [15, 18].

One key outcome of this analysis was the decline in revenue from drinks, in comparison to similar initiatives. A study in community sporting clubs found that implementing similar changes increased (self-reported) non sugar sweetened beverage sales in comparison to control clubs, without affecting overall revenue [18]. This may show that with implementation support and resources, the introduction of comprehensive healthy food and drink policies can improve the health of purchases and maintain revenue. Though not discussed, tap water consumption (reducing the purchase of bottled water) could account for the change in drink revenue, leading to potential sustainability benefits through reduced plastic waste, though consideration of other product lines which can make up lost revenue might mitigate revenue loss.

## Unanswered questions and future research

This study, along with the majority of previous healthy food and drink policy analyses, share two major limitations, the first being the inability to assess compensatory behaviours and substitution. Similarly, sales data cannot fully be used as a proxy for consumption, as customers can bring external purchases into centres. Innovative study designs are required to address

both of these limitations, especially when evaluating the effectiveness of policies in changing overall consumption patterns, and estimating effects on health [37]. Additionally, studies investigating healthy food and drink initiative implementation often report concerns of increased costs to retailers [38, 39]. Despite this, changes to food and operational costs as a result of improving the healthiness of items for sale, and how food costs can be optimised to increase the likelihood of switches to healthier menus being profitable, have not been investigated. This should also include exploration of acceptable, competitively priced options to replace unhealthy items, including packaged and preprepared items for retailers without full food preparation facilities. Successful strategies can then be incorporated into tools and training and used as an adjunct to implementation support practices.

## Conclusions

This evaluation of the introduction of a comprehensive healthy food and drink policy indicates an improvement in the healthiness of centre food environments following its introduction, as seen through reduced sales of 'red' food and drinks, and a reduction in both food and drink sugar contents and energy densities. This study adds to the evidence that policies to improve the healthiness of retail food environments can be effective and result in sustained improvements to the healthiness of customer purchases.

## Supporting information

**S1 Table. Change in healthiness of percentage sales ($) of food and drink between post and pre-implementation periods.** [¥] Pooled meta-analysis estimates using a random effects REML model. [#] % sales outcomes based on total volume sold. Mean centre marginal means estimated at each study period. $I^2$ estimate of the percentage of the between study variability. [&] Cochran's Q test for heterogeneity: $p < 0.001$ for all outcomes. CI confidence interval.
(DOCX)

**S1 Fig. Percentage difference[#] in total food and drink sales ($) between post and pre-implementation periods, stratified by centre characteristics.** a) Food by SEP, b) Food by type of preparation facilities, c) Drink by SEP and d) Drink by type of preparation facilities. # Relative percentage difference = (post–pre-implementation) / pre-implementation× 100. Overall effect estimated using pooled meta-analysis estimates using a random effects REML model. CI: confidence interval. SEP socioeconomic position of centre measured using the centre postcode and the Australian Bureau of Statistics Socio-Economic Indexes for Areas (SEIFA), Index of Relative Socio-Economic Advantage and Disadvantage (IRSAD) with percentiles for the state classified as being high SEP (IRSAD percentile >65, relative lack of disadvantage and greater advantage), medium (IRSAD percentile 34–65), and low (IRSAD percentile < 34, relative greater disadvantage and a lack of advantage).
(TIF)

**S2 Fig. Sympercentage difference[#] in total food and drink sales ($) between post and pre-implementation periods.** a) Food and b) Drink. [#] percentage difference on the 100 log(e) scale. Overall effect estimated using a random effects REML model. CI: confidence interval.
(TIF)

**S3 Fig. Percentage volume of 'red', 'amber', and 'green' food and drink monthly sales over time during the study period[#] by centre.** a) Food. [#] Dashed lines denote the date initiative started (1 January 2015) and date initiative was to be fully implemented (1 January 2017). b) Drink. [#] Dashed lines denote the date initiative started (1 January 2015) and date initiative was

to be fully implemented (1 January 2017).
(TIF)

**S4 Fig. Percentage volume of 'red', 'amber', and 'green' food and drink sold by study period and centre (mean and 95% CI).** FOOD: a) 'Red' food, b) 'Amber' food, c) 'Green' food. DRINK: d) 'Red' drink, e) 'Amber' drink, f) 'Green' drink. # Marginal means and 95% confidence intervals (CI) for each centre were estimated from a linear model with Newey-West standard errors to accommodate for serial autocorrelation (lag 3) and adjusting for calendar month and monthly attendance.
(TIF)

**S5 Fig. Difference[#] in percentage volume/ weight of 'red', 'amber', and 'green' food and drink sold between post and pre-implementation periods.** FOOD: a) 'Red' food, b) 'Amber' food, c) 'Green' food. DRINK: d) 'Red' drink, e) 'Amber' drink, and f) 'Green' drink. [#] Percentage difference = (post–pre-intervention) × 100. Overall effect estimated using pooled meta-analysis estimates using a random effects REML model. CI: confidence interval.
(TIF)

**S6 Fig. Nutrient[#] sales by study period and centre (means and 95% CI).** FOOD: a) Energy density (kJ/g), b) Percentage volume of sugar, c) Percentage volume of total fat, d) Percentage volume of saturated fat, and e) Percentage volume of salt. DRINK: f) Energy density (kJ/g), g) Percentage volume of sugar, h) Percentage volume of total fat, i) Percentage volume of saturated fat, and j) Percentage volume of salt. [#] Marginal means and 95% confidence intervals (CI) for each centre were estimated from a linear model with Newey-West standard errors to accommodate for serial autocorrelation (lag 3) and adjusting for calendar month and monthly attendance.
(TIF)

**S7 Fig. Difference[#] in nutritional outcomes between post and pre-implementation periods.** FOOD: a) Energy density (kJ/g), b) Percentage volume of sugar, c) Percentage volume of total fat, d) Percentage volume of saturated fat, e) Percentage volume of salt. DRINK: f) Energy density (kJ/g), g) Percentage volume of sugar, h) Percentage volume of total fat, i) Percentage volume of saturated fat, j) Percentage volume of salt. [#] Difference = (post–pre-intervention). Overall effect estimated using pooled meta-analysis estimates using a random effects REML model. CI: confidence interval.
(TIF)

## Author Contributions

**Conceptualization:** Shaan Stephanie Naughton, Anna Peeters, Alexandra Chung, Alethea Jerebine, Tara Boelsen-Robinson.

**Data curation:** Shaan Stephanie Naughton, Helena Romaniuk, Alethea Jerebine, Liliana Orellana, Tara Boelsen-Robinson.

**Formal analysis:** Shaan Stephanie Naughton, Helena Romaniuk, Anna Peeters, Alexandra Chung, Liliana Orellana, Tara Boelsen-Robinson.

**Funding acquisition:** Anna Peeters.

**Investigation:** Shaan Stephanie Naughton, Anna Peeters, Alexandra Chung, Tara Boelsen-Robinson.

**Methodology:** Shaan Stephanie Naughton, Helena Romaniuk, Anna Peeters, Alexandra Chung, Alethea Jerebine, Liliana Orellana, Tara Boelsen-Robinson.

**Project administration:** Shaan Stephanie Naughton, Alethea Jerebine, Tara Boelsen-Robinson.

**Resources:** Shaan Stephanie Naughton, Tara Boelsen-Robinson.

**Software:** Liliana Orellana.

**Supervision:** Anna Peeters, Liliana Orellana, Tara Boelsen-Robinson.

**Validation:** Shaan Stephanie Naughton, Helena Romaniuk, Anna Peeters, Alexandra Chung, Liliana Orellana, Tara Boelsen-Robinson.

**Visualization:** Shaan Stephanie Naughton, Helena Romaniuk, Anna Peeters, Liliana Orellana, Tara Boelsen-Robinson.

**Writing – original draft:** Shaan Stephanie Naughton, Helena Romaniuk, Anna Peeters, Liliana Orellana, Tara Boelsen-Robinson.

**Writing – review & editing:** Shaan Stephanie Naughton, Helena Romaniuk, Anna Peeters, Alexandra Chung, Alethea Jerebine, Liliana Orellana, Tara Boelsen-Robinson.

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
