## [Decision Letter · Decision Letter 0]

1 Feb 2023

PONE-D-22-25993Evaluation of the introduction of a healthy food and drink policy in 13 community recreation centres on the healthiness and nutrient content of customer purchases and business outcomes an observational study.PLOS ONE

Dear Dr. Naughton,

Thank you for submitting your manuscript to PLOS ONE. After careful consideration, we feel that it has merit but does not fully meet PLOS ONE’s publication criteria as it currently stands. Therefore, we invite you to submit a revised version of the manuscript that addresses the points raised during the review process.

The authors are advised to carefully address each of the reviewers' concerns and recommendations in a revised version of the manuscript and response to reviewers, with particular attention paid to the requested details regarding the data, methods, and policies, and with care regarding causal inference. Additionally, more detail regarding the statistical modelling would be of benefit in a revised version.

We look forward to receiving your revised manuscript.

Kind regards,

Blake Byron Walker, Ph.D.

Academic Editor

PLOS ONE

and https://journals.plos.org/plosone/s/file?id=ba62/PLOSOne_formatting_sample_title_authors_affiliations.pdf.

3. During the internal evaluation of the study, we have noted that the ethics approval number suggests that ethics approval for the study was obtained in 2016. As such of the data used for retrospective analysis was obtained in 2018, it is in our understanding that ethics approval should be obtained following this period. Please could you clarify whether the IRB approved for use of data prospective to the date of approval.

“This research was in part funded (to AP) by the National Health and Medical Research Council (NHMRC) as a  Centre of Research Excellence in Food Retail Environments for Health (RE-FRESH) (APP1152968)https://www.nhmrc.gov.au”

“SN, HR, AP, LO, and TBR are researchers within the NHMRC-funded Centre of Research Excellence in Food Retail Environments for Health (RE-FRESH) (APP1152968). SN is funded, and HR part funded, by RE-FRESH. AP is funded through an NHMRC Investigator Grant (APP1194630). AJ is funded through a Cotutelle Doctoral Studentship from Deakin University and Coventry University. TBR is funded through a Deakin University Postdoctoral Fellowship.

All authors have completed the ICMJE uniform disclosure form at www.icmje.org/coi_disclosure.pdf and declare: Anna Peeters receives funding from the NHMRC and is a board member of both Obesity Australia and the Victorian Health Promotion Foundation. Alexandra Chung receives funding from the Medical Research Future Fund Preventive Health Research Initiative (1199826). Liliana Orellana has receiving consulting fees from Western Health and the World Health Organisation- Fiji through payments made to Deakin University. Alethea Jerebine was an employee of YMCA Victoria 2014-2021. Shaan Naughton, Helena Romaniuk, and Tara Boelsen-Robinson report no other funding; All authors disclose no other relationships or activities that could appear to have influenced the submitted work.”

7. We note that you have indicated that data from this study are available upon request. PLOS only allows data to be available upon request if there are legal or ethical restrictions on sharing data publicly. For more information on unacceptable data access restrictions, please see http://journals.plos.org/plosone/s/data-availability#loc-unacceptable-data-access-restrictions.

Reviewers' comments:

Reviewer's Responses to Questions

**Comments to the Author**

1. Is the manuscript technically sound, and do the data support the conclusions?

Reviewer #1: No

Reviewer #2: Yes

Reviewer #3: Partly

Reviewer #4: Yes

2. Has the statistical analysis been performed appropriately and rigorously? 

Reviewer #1: No

Reviewer #2: Yes

Reviewer #3: Yes

Reviewer #4: Yes

3. Have the authors made all data underlying the findings in their manuscript fully available?

Reviewer #1: No

Reviewer #2: No

Reviewer #3: Yes

Reviewer #4: No

4. Is the manuscript presented in an intelligible fashion and written in standard English?

Reviewer #1: Yes

Reviewer #2: Yes

Reviewer #3: Yes

Reviewer #4: Yes

5. Review Comments to the Author

Reviewer #1: I appreciate the opportunity to review this article. The study compares sales of a three period public policy. The analysis needs to build a counterfactual to compare actual and expected sales. There are many possible alternatives to build this counterfactuals (depending on the data availability), such as, demand systems, time series models, panel data models. Another possibility, it is to have recreational community center without policy intervention (I am not sure whether it is is possible). Also, estimation needs to have robustness checks. Finally, the background and discussion sections needs to show the prolific evidence on food labelling (particularly, front of packaging). I do believe that the topic is interesting, however, the analysis needs to be taken further. Regards

Reviewer #2: I hope the following fairly minor thoughts are helpful:

Line 168 why not just say something like expressed as a “percentage difference”

Line 169 add x100%. And why not say the “difference” rather than “absolute mean difference” https://en.wikipedia.org/wiki/Absolute_difference

Such alterations would need to also be made in other places.

Line 232 delete “a reduction of”

Line 235 95% CI does not include 37.1%

Line 231 6 superscript should be [6]?

Line 393 “healthiness” not health

I wonder if Fig S2 should be in the main manuscript.

I wonder if Figures 1-3 should instead be supplementary, and a table with the overall results added to the main manuscript. (Table S1 could become part of this table.)

Describing outcome levels pre-implementation in the main manuscript (perhaps in a table) could help the reader appreciate the importance of changes of X.XX without having to look at suppl material. This comment is most relevant for: energy density, %vol sugar. Perhaps also for total fat, sat fat and sodium values.

use the word implementation consistently, not intervention nor initiative

inter-rater reliability was 93% … does this mean kappa =0.93?

meta-analysis better done with ratios rather than percent differences* … this might make a little difference to Figure 1a, I would expect -3.2 to get smaller

*unless use Tim Cole’s definition of a “symmetric percentage difference” (Stat Med 2000) in which case the two are equivalent. This paper might lead you to natural log-transform sales before then following your analysis approach.

how were SEs found for percent differences for an outcome for a centre? Using pre and post means and SEs somehow…

percent differences for total food and drink sales can be rounded to nearest integer

Reviewer #3: This paper studies how the introduction of a food labelling policy at 13 recreation centres in Victoria, Australia was associated with the sales of foods at these centres. The findings point to a shift to healthier food purchases over the study period. The study represents original research and therefore fits well within the scope of the journal. I have some concerns and suggestions which I detail in the comments below.

Comments

1. Causal identification

The key shortcoming of the study is that there is no credible control group for the policy intervention: basically, food sales before the implementation are compared to food sales after the implementation. For this difference to be interpretable as causal, any secular trends need to be absent. It is possible that there are general trends away from unhealthy foods and towards more healthful foods, and so food sales in 2018 can be different from those in 2013 for a variety of reasons unrelated to the specific food policy the paper considers.

For instance, the timing of the food policy in the paper largely coincides with other government food initiatives aimed at increasing the healthfulness of food, such as the roll-out of the Health Star Rating in supermarkets, which had an effect on the composition of the foods (e.g, Bablani et al. 2020, PLOS Medicine, 17(11): e1003427) and the purchases of consumers (e.g., Bablani et al. 2022, BMJ Nutrition, Prevention & Health, 5(2): 227–234). Therefore, changes in food sales in recreation centres might have been partly influenced by changes in patron’s preferences/diets over this period that were due to other policies such as the Health Star Rating, or simply because of other trends in the demands for foods over the study period.

I have two suggestions in this regard:

1) The paper should discuss some of these other broader food policies happening concurrently in Australia, such as the mentioned Health Star Rating, as they represent a salient potential confounding mechanism.

2) The paper needs to either (i) rewrite all text parts that make causal claims to reflect that the estimates represent associations rather than causal effects or (ii) find an alternative empirical strategy that makes it possible to control for common trends.

Clearly, option (ii) is significantly more involved than option (i). A possibility for option (ii) might be to exploit the fact that centres differed in the timing of the adoption of the policy, as mentioned in the text and quite strikingly visible in figures S2. This setting fits a staggered differences-in-differences approach, where centres which are not (yet) treated can serve as control observations for centres which are already treated.

2. Policy description

The policy is described in the Introduction and in the subsection “Study Design And Policy Description”. I was not clear on how the classification “red/amber/green” was displayed (e.g., on the products themselves or on the menus?). I also did not understand if during the “phased implementation” which allowed “stock of ‘red’ items to be sold, rather than immediately removed from sale” (p.4) the red/amber/green classification was displayed.

3. Analysis

It would make the analysis easier to understand if the paper could include an equation for the linear models estimated to obtain the centre-level estimates and an equation to show the random effects meta-analysis. The latter would be particularly helpful as the term ‘random effects’ is used slightly differently in different disciplines, so a more precise mathematical statement would open the article to a broader readership. It would also make any statistical assumptions behind the aggregation of the centre-level results more readily assessable.

It was not clear to me why the linear models included calendar months as a control variable. Perhaps I am missing something, but calendar months are not correlated with the key variables of interest (the three study periods) since each study period contains all calendar months by design.

Finally, the modelling approach for some outcomes requires the estimation of predicted means in levels and then the subsequent transformation of these absolute means (in levels) into relative means (in percentages). Conducting statistical inference on such ratios of estimated quantities is not entirely trivial, as their distribution is not normal. An alternative approach which the paper should conduct as a sensitivity analysis is to specify the outcome variable in logarithms. In that way, the estimated coefficients of the linear model directly have the (approximate) desired percentage-change interpretation and inference is standard as it is based on a single estimated coefficient.

Reviewer #4: This manuscript is a generally well-written report of a study that analyses 6 years of sales data comparing before and after a policy to reduce unhealthy food and drink availability in recreation centres was implemented. The study is interesting and is a useful addition to the literature. However, I do have some comments that I hope will help to improve the manuscript.

• Is this policy aiming to address childhood obesity? The introduction gives statistics on childhood obesity but not adults. If the policy is not specifically targeted, could you add some information on adults too?

• Does your analysis take secular trends into account? E.g. Were there price increases over the 6-year study period? Could this account for food sales not changing? Were sales expected to increase over the study period? In which case, no change in sales could be seen as a decrease in sales compared to what was expected.

• As I was reading the Methods, I was wondering if the included centres were typical of YMCA centres. I would include information on included centres (lines 201-205; lines 208-211) earlier on, in the Methods.

• You noted that 1012 products were sold over 6 years. I would be interested to know if the amount of choice varied across centres.

• What is monthly attendance? The number of people visiting each month?

• What did you define as ‘no’, ‘limited’ and ‘full’ food preparation facilities?

• Could you clarify why 2017-18 is considered post-intervention? Did centres reverse the policy? Or did it take 2016-17 for all centres to fully implement the policy?

• Why is time lag of 3 assumed and not tested?

• Why were 13 months of data excluded? And how did you decide that 5 other months of data were invalid?

• Were products classified with brand-specific nutrient information? It looks like it might not have been, and some products vary hugely in their composition.

• The diamond markers in your figures may be too big, as some centres end up not having 95% CIs visible and some values not crossing 0 look like they do (e.g. centre 11 in Fig 2a).

• It would be good to know how much this policy changed the foods and drinks that were available. What was the baseline offering in terms of % red, green and amber? What was it post-intervention? Did centres comply with the policy?

• Were any other initiatives (either at these centres or in Victoria in general) considered to have potentially impacted sales over the 4 years after the policy was implemented?

6. PLOS authors have the option to publish the peer review history of their article (what does this mean?). If published, this will include your full peer review and any attached files.

Reviewer #1: No

Reviewer #2: **Yes: **Mark D Chatfield

Reviewer #3: No

Reviewer #4: **Yes: **Amy Yau

---

## [Author Response · Author response to Decision Letter 0]

30 Mar 2023

Editor, comment 1: When submitting your revision, we need you to address these additional requirements.

and https://journals.plos.org/plosone/s/file?id=ba62/PLOSOne_formatting_sample_title_authors_affiliations.pdf.

Response: Thank you, we have ensured that our manuscript meets the PLOS ONE style requirements.

Editor, comment 2-. Please provide additional details regarding participant consent. In the ethics statement in the Methods and online submission information, please ensure that you have specified (1) whether consent was informed and (2) what type you obtained (for instance, written or verbal, and if verbal, how it was documented and witnessed). If your study included minors, state whether you obtained consent from parents or guardians. If the need for consent was waived by the ethics committee, please include this information.

Response: The section on participant consent has been updated to include “This study received ethical approval through the Deakin University human research ethics committee (2016-065) as a low risk study, and was conducted in accordance with the World Medical Association declaration of Helsinki [19].” Beginning on line 118, and “Individual level data was not accessed, data was accessed at a centre/retailer level. Staff at centres were involved only for facilitating auditing of food retail environments and completing questionnaire related to usual suppliers of food and beverages available for sale from the retail outlet. A representative of the organisation (the general manager of Advocacy and Health Promotion) was a named investigator/applicant on the ethics application and provided organisational consent.” beginning on line 121.

Editor, comment 3- During the internal evaluation of the study, we have noted that the ethics approval number suggests that ethics approval for the study was obtained in 2016. As such of the data used for retrospective analysis was obtained in 2018, it is in our understanding that ethics approval should be obtained following this period. Please could you clarify whether the IRB approved for use of data prospective to the date of approval. 

Response: Thank you for this query, to clarify we applied for an ethics modification in 2020 for an extension to the original ethics date by two years, that was approved. We are happy to supply copies of these documents if requested.

Editor, comment 4- Thank you for stating in your Funding Statement:

“This research was in part funded (to AP) by the National Health and Medical Research Council (NHMRC) as a Centre of Research Excellence in Food Retail Environments for Health (RE-FRESH) (APP1152968)https://www.nhmrc.gov.au”

Response: Thank you for the comment, though please note that there is another funding declaration above the quoted declaration. For clarification these have been amended to read “This research was funded (to AP) by the Medical Research Future Fund (MRFF) Boosting Preventative Health Research initiative : Diet and chronic disease prevention: supporting implementation of priority actions in the food and nutrition system ((MRFF Boosting Prevention via the Sax Institute) (2018–2021) https://www.health.gov.au/initiatives-and-programs/medical-research-future-fund) and by the National Health and Medical Research Council (NHMRC) as a Centre of Research Excellence in Food Retail Environments for Health (RE-FRESH) ((APP1152968) https://www.nhmrc.gov.au). The funders had no role in study design, data collection and analysis, decision to publish, or preparation of the manuscript. There was no additional external funding received for this study.”

Editor, comment 5- Thank you for stating the following in the Competing Interests section:

“SN, HR, AP, LO, and TBR are researchers within the NHMRC-funded Centre of Research Excellence in Food Retail Environments for Health (RE-FRESH) (APP1152968). SN is funded, and HR part funded, by RE-FRESH. AP is funded through an NHMRC Investigator Grant (APP1194630). AJ is funded through a Cotutelle Doctoral Studentship from Deakin University and Coventry University. TBR is funded through a Deakin University Postdoctoral Fellowship.

All authors have completed the ICMJE uniform disclosure form at www.icmje.org/coi_disclosure.pdf and declare: Anna Peeters receives funding from the NHMRC and is a board member of both Obesity Australia and the Victorian Health Promotion Foundation. Alexandra Chung receives funding from the Medical Research Future Fund Preventive Health Research Initiative (1199826). Liliana Orellana has receiving consulting fees from Western Health and the World Health Organisation- Fiji through payments made to Deakin University. Alethea Jerebine was an employee of YMCA Victoria 2014-2021. Shaan Naughton, Helena Romaniuk, and Tara Boelsen-Robinson report no other funding; All authors disclose no other relationships or activities that could appear to have influenced the submitted work.”

Response: Thank you, the competing interests section has been updated to read “SN, HR, AP, LO, and TBR are researchers within the NHMRC-funded Centre of Research Excellence in Food Retail Environments for Health (RE-FRESH) (APP1152968). SN is funded, and HR part funded, by RE-FRESH. AP is funded through an NHMRC Investigator Grant (APP1194630). AJ is funded through a Cotutelle Doctoral Studentship from Deakin University and Coventry University. TBR is funded through a Deakin University Postdoctoral Fellowship.

All authors have completed the ICMJE uniform disclosure form at www.icmje.org/coi_disclosure.pdf and declare: Anna Peeters receives funding from the NHMRC and is a board member of both Obesity Australia and the Victorian Health Promotion Foundation. Alexandra Chung receives funding from the Medical Research Future Fund Preventive Health Research Initiative (1199826). Liliana Orellana has receiving consulting fees from Western Health and the World Health Organisation- Fiji through payments made to Deakin University. Alethea Jerebine was an employee of YMCA Victoria 2014-2021. Shaan Naughton, Helena Romaniuk, and Tara Boelsen-Robinson report no other funding; All authors disclose no other relationships or activities that could appear to have influenced the submitted work. This does not alter our adherence to PLOS ONE policies on sharing data and materials”

Editor, comment 6- In your Data Availability statement, you have not specified where the minimal data set underlying the results described in your manuscript can be found. PLOS defines a study's minimal data set as the underlying data used to reach the conclusions drawn in the manuscript and any additional data required to replicate the reported study findings in their entirety. All PLOS journals require that the minimal data set be made fully available. For more information about our data policy, please see http://journals.plos.org/plosone/s/data-availability.

Response: Thank you for raising this. The analysis of retailer sales data in evaluating the effects of real-world health policy implementation is crucial to ensure that policies are sufficiently robust to change customer purchasing habits, though can still be implemented, especially by smaller, independent retailers who may face greater resourcing challenges (related to staff training, stocking healthier alternative products, and buying power) in policy implementation. To do this, access to data which is commercial in confidence (financial data) is required. In this field there is not a precedent for open sharing of data. Using the dataset from a systematic review (underway in our wider research group) of all food retail environment interventions that have analysed sales data from retailers, of the 54 publications since 2010, none have made their underly commercial in confidence data available open access. Some of these studies have been published in PLOS journals (see: 10.1371/journal.pmed.1003952; 10.1371/journal.pmed.1003715; 10.1371/journal.pmed.1003729; 10.1371/journal.pone.0204005; 10.1371/journal.pmed.1003714; 10.1371/journal.pmed.1004116), demonstrating this. 

In agreement with data availability statements used in similar studies previously published by PLOS journals we have arranged for an independent data custodian to be responsible for data access and we have updated our data availability statement to read “Data cannot be shared publicly because of the conditions of the agreement with the commercial collaborator. For purposes of verification and replication of the study’s findings, data can be accessed from the data custodian, the Associate Dean- Research, Faculty of Health, Deakin University, Australia, peter.enticott@deakin.edu.au for researchers who meet the criteria for access to confidential data.” 

Editor, comment 7- We note that you have indicated that data from this study are available upon request. PLOS only allows data to be available upon request if there are legal or ethical restrictions on sharing data publicly. For more information on unacceptable data access restrictions, please see http://journals.plos.org/plosone/s/data-availability#loc-unacceptable-data-access-restrictions.

Response: Please see response to Editor, comment 6.

Editor, comment 8 - Please review your reference list to ensure that it is complete and correct. If you have cited papers that have been retracted, please include the rationale for doing so in the manuscript text, or remove these references and replace them with relevant current references. Any changes to the reference list should be mentioned in the rebuttal letter that accompanies your revised manuscript. If you need to cite a retracted article, indicate the article’s retracted status in the References list and also include a citation and full reference for the retraction notice.

Response: Thank you, we have not cited any retracted papers. In response to the reviewer 4, comment 1, reference 2 “Australian Institute of Health and Welfare. Australia's children. no. CWS 69. Canberra: AIHW; 2020.” has been replaced with “Australian Institute of Health and Welfare. Overweight and obesity. Canberra: AIHW, 2022.”

Additionally, in response to reviewer comments the following references have been added: 

25. Cole T. Sympercents: symmetric percentage differences on the 100 loge scale simplify the presentation of log transformed data. Statistics in medicine. 2000;19(22):3109-25. (Reviewer 2, comment 11, and Reviewer 3, comment 6) 

30. Temple NJ. Front-of-package food labels: A narrative review. Appetite. 2020;144:104485. doi: https://doi.org/10.1016/j.appet.2019.104485. (Reviewer 1, comment 3)

31. Jones A, Thow AM, Ni Mhurchu C, Sacks G, Neal B. The performance and potential of the Australasian Health Star Rating system: a four-year review using the RE-AIM framework. Aust N Z J Public Health. 2019;43(4):355-65. Epub 2019/05/30. doi: 10.1111/1753-6405.12908. PubMed PMID: 31141289. (Reviewer 3, comment 2, and Reviewer 4, comment 13)

32. Sacks G, Robinson, E, for the Food-EPI Australia project team. Policies for tackling obesity and creating healthier food environments: 2019 Progress update, Victorian Government. Melbourne: Deakin University, 2019. (Reviewer 3, comment 2, and Reviewer 4, comment 13)

33. Sacks G, Robinson, E, for the Food-EPI Australia project team. Policies for tackling obesity and creating healthier food environments: 2019 Progress update, Australian Federal Government.

. Melbourne: Deakin University, 2019. (Reviewer 3, comment 2, and Reviewer 4, comment 13)

Reviewer 1, comment 1- I appreciate the opportunity to review this article. The study compares sales of a three period public policy. The analysis needs to build a counterfactual to compare actual and expected sales. There are many possible alternatives to build this counterfactuals (depending on the data availability), such as, demand systems, time series models, panel data models. Another possibility, it is to have recreational community center without policy intervention (I am not sure whether it is is possible). 

Response: Thank you for your comments. We acknowledge that the lack of control group is an important limitation in our study. A control group was not identified as the policy was to be implemented across all the facilities run by YMCA at the same time. There were a number of facilities that failed to implement the policy, however they could not be considered controls because 1) there was an extensive campaign within the organisation to improve food and drink healthiness that might have an effect on every centre; 2) the centres that didn’t implemented the policy are a self-selected sample and therefore not true controls. We have added in the limitations section a new paragraph emphasising these facts “The primary weakness of this study is its observational nature, lack of control centres, and the inability to account for the effect of other unmeasured factors including secular trends in sales or prices, and other initiatives implemented in Victoria or Australia. A control group could not be included as all centres in the state were required to implement the policy. Though there were a number of facilities that failed to implement the policy, these could not be considered controls as there was an extensive campaign within the organisation to improve food and drink healthiness that might have had an effect on every centre, and also as the centres that didn’t adhere to the policy are a self-selected sample and are therefore not true controls”– beginning on line 763. 

As explained in the statistical section we opted for comparing the mean outcome between periods without making the strong assumptions required by an interrupted time series analysis (which under a number of assumptions estimates the counterfactual change between periods) due to strong violations of model assumptions. Specifically, the linear trend assumption for each period was not sustained for many of the centres (see Figure S3). We have described in more detail this issue in the analysis section that now reads “The analytical approach and targets of estimation were chosen based on the following considerations: i) The change between post and pre initiative implementation was estimated at the centre level and the overall change calculated using a meta-analysis approach instead of jointly modelling each outcome for all centres, i.e., using a multilevel model. The main reason for this approach was that the seasonal sales patterns of different centres were not aligned (for example, some centres had peaks in sales in December, while others did not; see Figure S3); therefore, a multilevel model with a unique coefficient for each calendar month failed to adequately adjust for seasonal sales pattern at centre level. ii) The most common approach for the analysis of natural experiments interrupted time series analysis (ITSA), was found to not be appropriate for this analysis. ITSA estimates counterfactual outcomes (i.e., expected outcomes had the intervention not occurred) under a set of strong assumptions. We found that the linear assumption for the pre-implementation period was systematically violated for most outcomes in some of the centres making it impossible to estimate counterfactual outcomes (see Supplementary Fig. S3)” beginning line 319. We have further added this point to the limitations section “The primary weakness of this study is its observational nature, lack of control centres, and the inability to account for the effect of other unmeasured factors including secular trends in sales or prices, and other initiatives implemented in Victoria or Australia” beginning on line 763.

Reviewer 1, comment 2- Also, estimation needs to have robustness checks.

Response: We have performed the following robustness checks:

1) A sensitivity analysis on the impact of the time lag assumption. The original analysis assumed a time lag of 3. We have additionally considered time lags of 2, 4 and 5, and found that this assumption does not impact on the findings. The following sentence has been added: “To check the robustness of the model assumptions, the analysis was repeated with assumed lags of 2, 4 and 5, findings were similar across the different lag values (results not reported)” (beginning line 310).

2) For the total $ sales outcomes a sensitivity analysis was performed, with the outcomes loge-transformed prior to modelling and the percentage difference reported as symmetric percentages as suggested by Reviewer 2, comment 11, and Reviewer 3, comment 6. 

3) A sensitivity analysis was performed, with percentage of sales calculated using total sales ($) rather than volume/weight (“Sensitivity analysis with the outcomes loge-transformed prior to modelling, found that findings were similar with sympercent difference estimates larger in magnitude, especially for drink sales (Supplementary Fig. S2)” lines 453-455.

Reviewer 1, comment 3- Finally, the background and discussion sections needs to show the prolific evidence on food labelling (particularly, front of packaging). I do believe that the topic is interesting, however, the analysis needs to be taken further. Regards

Response: As this study used the traffic light system to classify items, rather than as a food labelling scheme to influence purchasing (though this was adopted in some centres), this has not been added to the introduction as this is believed to be outside of the scope of this paper. The role of traffic light labelling in influencing food choice has now been included in the discussion “One factor which may have influenced the results of this study is that some centres chose to display traffic light labelling on the items available for sale, which may have further influenced customer choice [31]. “ beginning on line 774. 

Reviewer 2, comment 1- I hope the following fairly minor thoughts are helpful:

Line 168 why not just say something like expressed as a “percentage difference”

And why not say the “difference” rather than “absolute mean difference” https://en.wikipedia.org/wiki/Absolute_difference

Such alterations would need to also be made in other places. 

Response: Thank you for this comment, following your suggestion, we now refer to the estimates as “difference” and “percentage difference”. The text and figures have been changed accordingly. 

Reviewer 2, comment 2- Line 169 add x100%. 

Response: Thank you, this text, now on line 244 has been updated to read “For total food and drink sales ($), we estimated the percentage difference between post and pre-implementation sales, i.e. (mean sales post – mean sales pre)/mean sales pre × 100.” for clarity.

3- Line 232 delete “a reduction of” 

Response: Thank you, this text, now on line 467 has been updated to Reviewer 2, comment read “For `red’ food, the mean difference between post-implementation and pre-implementation sales was 15% (95% CI -22% to -7.7%, p <0.001), ‘amber’ food increased by 11% (95% CI 5.5% to 16%, p<0.001), and ‘green’ did not change (3.3%, 95% CI -1.4% to 8.0%, p=0.166) (Table 1, Fig. S5a-c).”.

Reviewer 2, comment 4- Line 235 95% CI does not include 37.1% 

Response: Thank you, this text, now on line 471 has been updated to read “The mean difference in ‘red’ drinks sold was -37% (95% CI -43% to -31%, p<0.001); ‘amber’ drinks increased by 8.8% (95% CI 3.6% to 14%, p=0.001), and ‘green’ drinks increased by 28% (95% CI 23% to 33%, p<0.001) (Fig. S5d-f).” as the ‘declined’ before the 37.1% indicate a change of -37.1%, within the 95% CI, the sentence has been corrected to refer to a change of -37.1%. 

Reviewer 2, comment 5- Line 231 6 superscript should be [6]? 

Response: Thank you, the reference style has been corrected.

Reviewer 2, comment 6- Line 393 “healthiness” not health 

Response: This has been corrected, now on line 920.

Reviewer 2, comment 7- I wonder if Fig S2 should be in the main manuscript.

I wonder if Figures 1-3 should instead be supplementary, and a table with the overall results added to the main manuscript. (Table S1 could become part of this table.) 

Response: Thank you for this suggestion. Figure 2 and 3 have now been moved to the supplemental data, and we have modified Table 1 to include information on all outcomes except for sales outcomes.

Reviewer 2, comment 8- Describing outcome levels pre-implementation in the main manuscript (perhaps in a table) could help the reader appreciate the importance of changes of X.XX without having to look at suppl material. This comment is most relevant for: energy density, %vol sugar. Perhaps also for total fat, sat fat and sodium values. 

Response: Thank you for this suggestion. We have modified Table 1 which now includes information on all outcomes except for sales outcomes.

Reviewer 2, comment 9- use the word implementation consistently, not intervention nor initiative 

Response: Where “intervention” has been used to describe the study this has been replaced with ‘initiative’. In instances where the study periods have inadvertently been referred to as ‘…intervention’ these have been corrected to state ‘…implementation’. Changes have been introduced across the text, and tables and figures footnotes. 

Reviewer 2, comment 10- inter-rater reliability was 93% … does this mean kappa =0.93? 

Response: Thank you for this comment, the inter-rater reliability is percentage agreement. In response to this comment we have updated the manuscript to show kappa “There was good inter-rater reliability for categorisation of food items (kappa 0.83).” beginning line 418. 

Reviewer 2, comment 11- meta-analysis better done with ratios rather than percent differences* … this might make a little difference to Figure 1a, I would expect -3.2 to get smaller

*unless use Tim Cole’s definition of a “symmetric percentage difference” (Stat Med 2000) in which case the two are equivalent. This paper might lead you to natural log-transform sales before then following your analysis approach. 

Response: Thank you for this suggestion. We have estimated symmetric percentage differences using Tim Cole’s definition (2000) and have reported these results as a sensitivity analysis. The estimates changed slightly but conclusions remained the same. Please note, in our data, there is obvious temporal ordering which according to Cole (2000) is a case that justifies estimating the percentage difference using the conventional approach. 

New text added: “For the total $ sales outcomes a sensitivity analysis was performed, fitting the same model with outcomes loge-transformed; and the percentage difference reported as sympercents [25].” (lines 246-248) “Sensitivity analysis with the outcomes loge-transformed prior to modelling, found that findings were similar with sympercent difference estimates larger in magnitude, especially for drink sales (Supplementary Fig. S2). ” (lines 453-455) 

Reviewer 2, comment 12- how were SEs found for percent differences for an outcome for a centre? Using pre and post means and SEs somehow… 

Response: SEs for the percent differences were estimated using the Stata command nlcom, which uses the delta method to estimate the standard error of non-linear combinations of the estimated parameters.

Reviewer 2, comment 13- percent differences for total food and drink sales can be rounded to nearest integer 

Response: We have edited all estimates to have two significant figures.

Reviewer 3, comment 1- This paper studies how the introduction of a food labelling policy at 13 recreation centres in Victoria, Australia was associated with the sales of foods at these centres. The findings point to a shift to healthier food purchases over the study period. The study represents original research and therefore fits well within the scope of the journal. I have some concerns and suggestions which I detail in the comments below. Thank you

Reviewer 3, comment 2- Causal identification

The key shortcoming of the study is that there is no credible control group for the policy intervention: basically, food sales before the implementation are compared to food sales after the implementation. For this difference to be interpretable as causal, any secular trends need to be absent. It is possible that there are general trends away from unhealthy foods and towards more healthful foods, and so food sales in 2018 can be different from those in 2013 for a variety of reasons unrelated to the specific food policy the paper considers.

For instance, the timing of the food policy in the paper largely coincides with other government food initiatives aimed at increasing the healthfulness of food, such as the roll-out of the Health Star Rating in supermarkets, which had an effect on the composition of the foods (e.g, Bablani et al. 2020, PLOS Medicine, 17(11): e1003427) and the purchases of consumers (e.g., Bablani et al. 2022, BMJ Nutrition, Prevention & Health, 5(2): 227–234). Therefore, changes in food sales in recreation centres might have been partly influenced by changes in patron’s preferences/diets over this period that were due to other policies such as the Health Star Rating, or simply because of other trends in the demands for foods over the study period.

I have two suggestions in this regard:

1) The paper should discuss some of these other broader food policies happening concurrently in Australia, such as the mentioned Health Star Rating, as they represent a salient potential confounding mechanism.

2) The paper needs to either (i) rewrite all text parts that make causal claims to reflect that the estimates represent associations rather than causal effects or (ii) find an alternative empirical strategy that makes it possible to control for common trends.

Clearly, option (ii) is significantly more involved than option (i). A possibility for option (ii) might be to exploit the fact that centres differed in the timing of the adoption of the policy, as mentioned in the text and quite strikingly visible in figures S2. This setting fits a staggered differences-in-differences approach, where centres which are not (yet) treated can serve as control observations for centres which are already treated. 

Response: Thank you for this comment. The lack of a control group has been further explained in the response to Reviewer 1, comment 1.

In regard to secular trends, the present policy implementation was the only initiative implemented at the YMCA centres during the implementation and post-implementation periods. No other policies relating to sports and recreation settings were enacted during this period, with YMCA Victoria being the first organisation to introduce the voluntary policy across all of their centres. In all general Victorian settings, the only policy to be introduced was the introduction of the same voluntary policy in schools, childcare, and workplaces. The only other main general policy which may have had an influence on the provision of healthier food is the introduction of the Health Star Rating system in 2014, though an evaluation of the system, in October 2018, found that population awareness of the scheme was low, and that only 20-28% of eligible products adopted the rating system, with this being skewed towards items that received a higher rating (higher rating indicates a healthier alternative when comparing like products). The limitations of this study have been updated to include this reading “The primary weakness of this study is its observational nature, lack of control centres, and the inability to account for the effect of other unmeasured factors including secular trends in sales or prices, and other initiatives implemented in Victoria or Australia. A control group could not be included as all centres in the state were required to implement the policy. Though there were a number of facilities that failed to implement the policy, these could not be considered controls as there was an extensive campaign within the organisation to improve food and drink healthiness that might have had an effect on every centre, and also as the centres that didn’t adhere to the policy are a self-selected sample and are therefore not true controls. One factor which may have influenced the results of this study is that some centres chose to display traffic light labelling on the items available for sale, which may have further influenced customer choice [31]. Other healthy eating policies in place at the time may have influenced secular trends, though healthy food policy progress was limited in Victoria [32], and federally [33], at the time. No other policies relating to sports and recreation settings were enacted during this period, with this organisation being the first to introduce the voluntary policy across all of their centres [9]. At a state level, the only policy to be introduced was the introduction of the same voluntary policy in schools, childcare, and workplaces [32]. The only other policy which may have had an influence on the consumption of healthier food was the introduction of the Health Star Rating system in 2014, though an evaluation of the system, in October 2018, found that population awareness of the scheme was low, and that only 20-28% of eligible products adopted the rating system, with this being skewed towards items that received a higher rating (higher rating indicates a healthier alternative when comparing like products) [34]. Also, as can be seen from the outcomes plotted over time, changes to sales occurred at different times as the policy was implemented, decreasing the likelihood of external extraneous factors being the primary factor influencing results. Another limitation of this study was the number of centres eligible for inclusion (37% of non-seasonal centres) in the analysis. Though centre eligibility may have been influenced by the length of the study, investigation of long-term outcomes was key. The change to the salt content of food purchases may also have been underestimated, as several ‘red’ items that were removed from sale often have additional salt added after cooking (e.g., deep fried foods), which is not included in the nutritional information. Finally, not all items sold could be classified, with the majority of items excluded from the volume and nutritional analysis being hot beverages. ”, beginning line 763.

2) Regarding the second part of this comment:(i) We have amended the text to reflect that we are estimating associations rather than causal effects.

ii) Unfortunately, we were not able to find an alternative strategy that made it possible to control for common trends. A control group was not included in the study design, and retrospective data collection would not be feasible. 

At the end of 2014, YMCA introduced a policy for centres to implement ‘Healthy Choices guidelines’ which were expected to be in place by the end of December 2016. 

A staggered difference-in-difference approach was considered and ruled out. During the implementation period, each centre was responsible for making their own changes, both in the food and drink available and in the timing of these changes. We do not have detailed centre records to identify when they started the intervention, and it would be problematic to use sales data (the outcome measures) for this purpose. 

As the implementation of the policy and the start date varied across facilities, we chose to perform an intention to treat analysis, with start dates for each phase defined by policy milestones as set by YMCA Victoria. Therefore, we assessed the “impact” of the policy without considering its level of implementation. This is the main reason we only present comparisons between the pre- and post- implementation sales outcomes.

For more detail on the lack of control group and analytical approach, please see our response to Reviewer 1, comment 1.

Reviewer 3, comment 3- Policy description

The policy is described in the Introduction and in the subsection “Study Design And Policy Description”. I was not clear on how the classification “red/amber/green” was displayed (e.g., on the products themselves or on the menus?). I also did not understand if during the “phased implementation” which allowed “stock of ‘red’ items to be sold, rather than immediately removed from sale” (p.4) the red/amber/green classification was displayed. 

Response: Thank you for this comment, the subsection “Study design and policy description” has been updated to include “The policy stipulated that of the items available for purchase <10% of items were to be ‘red’ classified, and >50% ‘green’. Traffic light labelling of items for sale was not explicitly in the policy and was at the discretion of the individual centres.” beginning line 139.

Reviewer 3, comment 4- Analysis

It would make the analysis easier to understand if the paper could include an equation for the linear models estimated to obtain the centre-level estimates and an equation to show the random effects meta-analysis. The latter would be particularly helpful as the term ‘random effects’ is used slightly differently in different disciplines, so a more precise mathematical statement would open the article to a broader readership. It would also make any statistical assumptions behind the aggregation of the centre-level results more readily assessable. 

Response: We have included the model equations in the paper (lines 228-241) “First, we estimated the change in the outcomes between post- and pre-policy implementation periods separately for each centre and outcome by fitting a linear model with Newey-West standard errors to accommodate for serial autocorrelation (time lag of 3 assumed). For each sales outcome Y, the model displayed in the following equation was fitted. 

Y_it =〖 β〗_i0+β_i1 t+I(24<t≤48) β_i2++I(48<t≤72) β_i3+β_i4 A_it+β_i5 M_(1,it)+β_i6 M_(2,it)+〖 β〗_i7 M_(3,it)+β_i8 M_(4,it)+β_i9 M_(5,it)+β_i10 M_(6,it)+ β_i11 M_(7,it)+β_i12 M_(8,it)+β_i13 M_(10,it)+β_i14 M_(11,it)+ β_i15 M_(12,it)+υ_it 

here Y_it represents an outcome for site i (i=1,2,…,13) at time t (t=1,2,3,…,72 months); I(B) is an indicator function representing the study period taking the value 1 if condition B is true and 0 otherwise with pre-implementation period I(t≤24) used as the reference category; A_it represents monthly centre attendance at site i and time t; M_(1,it) to M_(12,it) are indicator variables for calendar month with September M_(9,it) used as the reference category, while the random variable υ_it represents the residual for centre i at time t.”

Reviewer 3, comment 5- It was not clear to me why the linear models included calendar months as a control variable. Perhaps I am missing something, but calendar months are not correlated with the key variables of interest (the three study periods) since each study period contains all calendar months by design. 

Response: Calendar months were included in the models as control variables to account for outcome variability in sales across the calendar year (See Fig S3). 

Reviewer 3, comment 6- Finally, the modelling approach for some outcomes requires the estimation of predicted means in levels and then the subsequent transformation of these absolute means (in levels) into relative means (in percentages). Conducting statistical inference on such ratios of estimated quantities is not entirely trivial, as their distribution is not normal. An alternative approach which the paper should conduct as a sensitivity analysis is to specify the outcome variable in logarithms. In that way, the estimated coefficients of the linear model directly have the (approximate) desired percentage-change interpretation and inference is standard as it is based on a single estimated coefficient. 

Response: We have performed the sensitivity analysis requested. Reviewer 2 also requested an alternative approach; we have responded in full above. Please see response to Reviewer 2, comment 11. 

Reviewer 4, comment 1- This manuscript is a generally well-written report of a study that analyses 6 years of sales data comparing before and after a policy to reduce unhealthy food and drink availability in recreation centres was implemented. The study is interesting and is a useful addition to the literature. However, I do have some comments that I hope will help to improve the manuscript.

• Is this policy aiming to address childhood obesity? The introduction gives statistics on childhood obesity but not adults. If the policy is not specifically targeted, could you add some information on adults too? 

Response: Thank you for this feedback and the suggestion of including adult obesity rates. This sentence has been updated to include adult prevalence of overweight and obesity, reading “With approximately 1 in 4 Australian children (aged 5–14) and 2/3rds of adults experiencing overweight or obesity in 2018 [2], there is now a concerted focus on improving the health of environments where foods are advertised, purchased, and consumed [3].” and the reference (2) in this sentence changed, beginning line 71.

Reviewer 4, comment 2- Does your analysis take secular trends into account? E.g. Were there price increases over the 6-year study period? Could this account for food sales not changing? Were sales expected to increase over the study period? In which case, no change in sales could be seen as a decrease in sales compared to what was expected. 

Response: Thank you for this comment. We were unable to explore price increases over the 6-year study period as aggregate data shows the average price paid per item for each month, and very few products were sold for the entire 6 years. Additionally, revenue from food service was not considered a major income source by the managing organisation, so it is unlikely they would increase prices as a profit generating mechanism. Sales were not expected to increase over the study period, and centres moving to larger facilities, which may’ve increased attendance, and therefore potentially sales, were excluded from the study. We have updated the limitations of the study, beginning line 763 to include this “The primary weakness of this study is its observational nature, lack of control centres, and the inability to account for the effect of other unmeasured factors including secular trends in sales or prices, and other initiatives implemented in Victoria or Australia.”

Reviewer 4, comment 3- As I was reading the Methods, I was wondering if the included centres were typical of YMCA centres. I would include information on included centres (lines 201-205; lines 208-211) earlier on, in the Methods. 

Response: Thank you for this comment, though as the centre characteristics are a result of applying the inclusion criteria to the available centres, we believe it is appropriately presented at the beginning of the results section.

Reviewer 4, comment 4- You noted that 1012 products were sold over 6 years. I would be interested to know if the amount of choice varied across centres. 

Response: Thank you for this comment, though sales data does not necessarily reflect item availability (as items may have been available though didn’t sell). The statement “The sales data indicates that the amount of choice varied across centres, with the range of individual items available over the study period being 110-467, with a median of 273.” has been added to the manuscript beginning on line 426.

Reviewer 4, comment 5- What is monthly attendance? The number of people visiting each month? 

Response: This sentence, beginning line 199 has been updated to read “Monthly attendance data, number of people entering centre through turnstiles (automatically counted), was obtained through YMCA’s central database.”. 

Reviewer 4, comment 6- What did you define as ‘no’, ‘limited’ and ‘full’ food preparation facilities?

Response: Thank you, this information beginning on line 205 manuscript has been updated to read “Type of food preparation at each centre was classified as: no facilities (no preparation facilities, limited storage (fridge), ability to heat packaged and/or preprepared items); limited facilities (ability to prepare simple items (e.g. sandwiches) and heat packaged and/or preprepared items, limited storage (fridge)); or full food preparation facilities (ability to prepare and serve a range of hot and cold meals (including to order), full storage (ambient/cold/frozen)).”

Reviewer 4, comment 7- Could you clarify why 2017-18 is considered post-intervention? Did centres reverse the policy? Or did it take 2016-17 for all centres to fully implement the policy? 

Response: As described in the methods, centres gradually introduced the policy over the 2016-2017 implementation period. Erroneous use of ‘post intervention’ in the manuscript has been corrected to read ‘post-implementation’. Therefore 2017-2018 is considered post- implementation.

Reviewer 4, comment 8-Why is time lag of 3 assumed and not tested? 

Response: We assumed a time lag of 3, we have found and reported in the paper that time lags of 2, 4 and 5 do not impact our findings. Please see response to Reviewer 1, comment 2 (robustness checks) for more detail. 

Reviewer 4, comment 9- Why were 13 months of data excluded? And how did you decide that 5 other months of data were invalid? 

Response: We have amended the text to expand on why data was excluded or considered invalid “Thirteen sequential months of sales data were excluded (1.4%: first 4 months for 1 centre as it was opening up a new café, and first 9 months in another centre that did not provide complete sale data), and 5 non-sequential months in 2 different centres with partially incomplete sales data (0.5%) had their outcomes replaced with previous months’ values carried forward for analysis.” beginning line 410. 

Reviewer 4, comment 10- Were products classified with brand-specific nutrient information? It looks like it might not have been, and some products vary hugely in their composition. 

Response: Where brand specific descriptors were provided, or brands could be identified from standard ordering questionnaires completed at a store level, brand specific nutrient information was used. When this was not available an average of the nutritional composition of available brands for the same product was used. We have updated the results to include the statement beginning line 423 “78% of unique items were packaged items coded with brand specific nutrient information, with the composition of 6.3% calculated from recipes provided or standard serves. Items that had their composition estimated from point of sale information accounted for 0.83% of the total items sold over the study period.” 

Reviewer 4, comment 11- The diamond markers in your figures may be too big, as some centres end up not having 95% CIs visible and some values not crossing 0 look like they do (e.g. centre 11 in Fig 2a). 

Response: Thank you for pointing this out. We have changed all forest plots using smaller markers and, in some plots, changing the scale.

Reviewer 4, comment 12- It would be good to know how much this policy changed the foods and drinks that were available. What was the baseline offering in terms of % red, green and amber? What was it post-intervention? Did centres comply with the policy? 

Response: Centre compliance was monitored by individual centre managers and reported to the managing organisation. Comparing this compliance check data to sales data identified discrepancies between the two, especially in relation to the sale of red items. As such an intention to treat approach was taken as we were not able to independently verify compliance, or that the classification of items done by centre staff, was correct. As only sales data was available for analysis, we are not able to determine what was available for sale, only what was purchased. 

Reviewer 4, comment 13- Were any other initiatives (either at these centres or in Victoria in general) considered to have potentially impacted sales over the 4 years after the policy was implemented?

Response: Thank you for this comment, other initiatives are discussed in response to Reviewer 3, comment 2.

---

## [Decision Letter · Decision Letter 1]

4 Jul 2023

Evaluation of the introduction of a healthy food and drink policy in 13 community recreation centres on the healthiness and nutrient content of customer purchases and business outcomes: an observational study.

PONE-D-22-25993R1

Dear Dr. Naughton,

We’re pleased to inform you that your manuscript has been judged scientifically suitable for publication and will be formally accepted for publication once it meets all outstanding technical requirements.

Kind regards,

Blake Byron Walker, Ph.D.

Academic Editor

PLOS ONE

Additional Editor Comments (optional):

Reviewers' comments:

Reviewer's Responses to Questions

**Comments to the Author**

1. If the authors have adequately addressed your comments raised in a previous round of review and you feel that this manuscript is now acceptable for publication, you may indicate that here to bypass the “Comments to the Author” section, enter your conflict of interest statement in the “Confidential to Editor” section, and submit your "Accept" recommendation.

Reviewer #2: All comments have been addressed

Reviewer #3: All comments have been addressed

Reviewer #4: All comments have been addressed

2. Is the manuscript technically sound, and do the data support the conclusions?

Reviewer #2: (No Response)

Reviewer #3: Yes

Reviewer #4: Yes

3. Has the statistical analysis been performed appropriately and rigorously? 

Reviewer #2: (No Response)

Reviewer #3: Yes

Reviewer #4: Yes

4. Have the authors made all data underlying the findings in their manuscript fully available?

Reviewer #2: (No Response)

Reviewer #3: Yes

Reviewer #4: No

5. Is the manuscript presented in an intelligible fashion and written in standard English?

Reviewer #2: (No Response)

Reviewer #3: Yes

Reviewer #4: Yes

6. Review Comments to the Author

Reviewer #2: (No Response)

Reviewer #3: (No Response)

Reviewer #4: I thank the authors for thoroughly addressing my comments. I have no further comments and recommend this manuscript for publication.

7. PLOS authors have the option to publish the peer review history of their article (what does this mean?). If published, this will include your full peer review and any attached files.

Reviewer #2: No

Reviewer #3: No

Reviewer #4: No

---

## [Editor Report · Acceptance letter]

7 Jul 2023

PONE-D-22-25993R1 

Evaluation of the introduction of a healthy food and drink policy in 13 community recreation centres on the healthiness and nutrient content of customer purchases and business outcomes: an observational study. 

Dear Dr. Naughton:

I'm pleased to inform you that your manuscript has been deemed suitable for publication in PLOS ONE. Congratulations! Your manuscript is now with our production department. 

Kind regards, 

on behalf of

Prof. Dr. Blake Byron Walker 

Academic Editor

PLOS ONE